# Divergent Mechanisms of H2AZ.1 and H2AZ.2 in PRC1-Mediated H2A Ubiquitination

**DOI:** 10.3390/cells14151133

**Published:** 2025-07-23

**Authors:** Xiangyu Shen, Chunxu Chen, Amanda E. Jones, Xiaokun Jian, Gengsheng Cao, Hengbin Wang

**Affiliations:** 1School of Life Sciences, Henan University, Kaifeng 475004, Chinajxk8097@gmail.com (X.J.);; 2Department of Internal Medicine, Division of Hematology, Oncology and Palliative Care, Massey Comprehensive Cancer Center, Virginia Commonwealth University, Richmond, VA 23298, USA; 3Department of Cell Biology, University of Texas Southwestern Medical Center, Dallas, TX 75390, USA

**Keywords:** histone variant, H2AZ.1, H2AZ.2, PRC1, H2AK119ub

## Abstract

The histone H2A variant H2AZ plays pivotal roles in shaping chromatin architecture and regulating gene expression. We recently identified H2AZ.2 in histone H2A lysine 119 ubiquitination (H2AK119ub)-enriched nucleosomes, but it is not known whether its highly related isoform H2AZ.1 also regulates this modification. In this study, we employed isoform-specific epitope-tagged knock-in mouse embryonic stem cell (ESC) lines to dissect the roles of each isoform in Polycomb Repressive Complex 1 (PRC1)-mediated H2AK119ub. Our results show that H2AZ.1 and H2AZ.2 share highly overlapping genomic binding profiles, both co-localizing extensively with H2AK119ub-enriched loci. The knockdown of either isoform led to reduced H2AK119ub levels; however, the two isoforms appear to function through distinct mechanisms. H2AZ.1 facilitates the recruitment of Ring1B, the catalytic subunit of PRC1, thereby promoting the deposition of H2AK119ub. In contrast, H2AZ.2 does not significantly affect Ring1B recruitment but instead functions as a structural component that stabilizes H2AK119ub-modified nucleosomes. In vitro ubiquitination assays indicate that H2AZ.1-containing nucleosomes serve as more efficient substrates for PRC1-mediated ubiquitination compared to those containing H2AZ.2. Thus, these findings define the distinct mechanisms of the two H2AZ variants in regulated PRC1-mediated H2AK119 ubiquitination and highlight a functional division of labor in epigenetic regulation.

## 1. Introduction

The dynamic organization and domain-specific functions of eukaryotic chromatin are orchestrated by a range of regulatory mechanisms, among which the incorporation of histone variants plays a pivotal role [1,2,3,4]. Unlike canonical histones, these variants exhibit unique sequence and structural features and are deposited into chromatin independent of DNA replication, typically outside the S phase of the cell cycle [5,6,7]. The targeted deposition at specific genomic loci reshapes local chromatin architecture, modulates nucleosome accessibility, and elicits distinct downstream outcomes. These processes contribute to the plasticity and specificity of spatially defined chromatin regulation [4,8].

H2AZ is one of the most functionally versatile histone variants, playing key roles in chromatin dynamics and gene regulation [9,10]. In *Saccharomyces cerevisiae*, *Caenorhabditis elegans*, and *Drosophila melanogaster*, a single H2AZ-encoding gene has been identified: *HTZ1*, *HTZ-1*, and *His2av*, respectively [11,12,13]. These gene products are thought to represent the ancestral function of H2AZ, and the encoded protein sequences are more similar to the H2AZ.2 isoform in vertebrates, suggesting a more conserved and essential function in chromatin regulation across eukaryotes [14,15]. Notably, *His2av* in *Drosophila* appears to fulfill the dual functions of both H2A.Z and H2A.X, highlighting its multifunctional nature [16,17]. In vertebrates, a gene duplication event gave rise to two closely related paralogous genes, *H2afz* and *H2afv*, which encode H2AZ.1 and H2AZ.2, respectively [18]. H2AZ.2 further undergoes alternative splicing to produce two variants, H2AZ.2.1 and H2AZ.2.2, the latter of which has a truncated C-terminal tail, potentially affecting protein–protein interactions within the nucleosome [19]. Despite isoform divergence, H2AZ retains conserved chromatin-regulatory functions across species [18,20]. H2AZ is widely known for its role in transcriptional regulation. Its incorporation into nucleosomes near promoter regions, particularly at + 1 nucleosome, facilitates transcription initiation and poising by enhancing promoter accessibility and maintaining genes in a state ready for rapid induction [4,10,21]. Moreover, H2AZ is required for DNA double-strand breaks and transcription-coupled repair, prevents aberrant transcription initiation, and maintains nucleosome boundary, thereby contributing to genome integrity [22,23]. While these functions are conserved, spanning from yeast to mammals, it remains unclear whether these diverse regulatory functions are shared between H2AZ.1 and H2AZ.2 in vertebrates or whether each isoform plays a distinct role in chromatin regulation.

H2AZ.1 and H2AZ.2 differ by only three amino acid residues, yet accumulating evidence suggests that these isoforms have evolved distinct and non-redundant functions across mammalian systems [14,24,25]. H2AZ.2 plays a critical role in maintaining genome stability by regulating chromosome segregation, facilitating chromatin reorganization in response to DNA damage, and promoting melanoma cell proliferation and drug responsiveness [26,27,28]. In contrast, H2AZ.1 is essential for cell proliferation and the transcriptional activation of cell cycle genes [29]. It is also aberrantly overexpressed in hepatocellular carcinoma, contributing to the organization of pericentric heterochromatin through its interaction with HP1β [30]. A key residue at position 38 (serine in H2AZ.1, threonine in H2AZ.2) has been linked to distinct protein–protein interactions and nucleosome stability and is thought to play a critical role in the pathogenesis of Floating–Harbor syndrome [31].

Previous studies have also shown that H2AZ is enriched at Polycomb target genes and often co-localizes with repressive histone marks such as H3K27me3 in mouse embryonic stem cells (mESCs) [32]. Furthermore, the monoubiquitination of H2AZ has been associated with facultative heterochromatin [33], suggesting a potential functional interaction with Polycomb repressive complexes. However, these studies did not distinguish between H2AZ.1 and H2AZ.2 isoforms. To address this question, two studies have attempted to dissect isoform-specific roles, one in human ESCs and one in human U2OS cells, using the knock-in strategy [15,31]. In this study, we developed a dual-tagged mESC line that expresses epitope-tagged H2AZ.1 and H2AZ.2 within the same cell line. This system enables a direct comparison of isoform-specific chromatin binding profiles and functional outputs under physiological conditions. Using this platform and through a combination of CUT & Tag, isoform-specific knockdown, and in vitro ubiquitination assays, we demonstrate that while the isoforms share similar genomic profiles; they regulate H2AK119ub through distinct mechanisms: H2AZ.1 facilitates PRC1 recruitment, whereas H2AZ.2 maintains the stability of H2AK119ub-marked nucleosomes. In vitro ubiquitination assays reveal that H2AZ.1-containing nucleosomes serve as more efficient substrates for PRC1-mediated ubiquitination compared to those containing H2AZ.2. Collectively, these findings demonstrate that H2AZ.1 and H2AZ.2 exert distinct, non-redundant functions in PRC1-mediated chromatin regulation.

## 2. Materials and Methods

### 2.1. Cells and Cell Cultures

Human embryonic kidney 293T (HEK293T, CRL-3216) and mESC line R1 (SCRC-1011) were obtained from ATCC (Manassas, VA, USA). mESC line V6.5 was described previously [34]. HEK293T cells were cultured in DMEM (MT010–013-CM, Corning Inc., Corning, NY, USA) supplemented with 10% fetal bovine serum (FBS; S11550, R & D Systems, Minneapolis, MN, USA) and 50 U/mL penicillin-streptomycin (15070063, Gibco, Grand Island, NY, USA) at 37 °C in a humidified incubator with 5% CO_2_. mESCs were maintained in ESC medium consisting of DMEM supplemented with 50 U/mL penicillin-streptomycin, 15% defined murine ESC FBS (S10250, R & D Systems, Minneapolis, MN, USA), 1% GlutaMAX (35050–061, Gibco, Grand Island, NY, USA), 1× non-essential amino acids (MT-25–025-CI, Corning, NY, USA), 1× nucleosides (ES-008-D, Millipore Burlington, MA, USA), 0.007% β-mercaptoethanol (M3148, Sigma-Aldrich, St. Louis, MO, USA), and 1000 U/mL mLIF (ESGRO, Millipore, Burlington, MA, USA) at 37 °C in a humidified 5% CO_2_ incubator. ESCs were cultured on irradiated mouse embryonic fibroblasts (MEFs; A34181, Fisherbrand, Waltham, MA, USA) or 0.1% gelatin-coated plate. Flag-H2AZ.2 knock-in mESCs and Flag-H2AZ.2 overexpressing 293T cell lines were described previously [35]. Flag-H2AZ.1-expressing 293 T cells were constructed following the procedure described in previous studies [35]. Flag-Ring 1B mESCs were obtained from Dr. Wang’s laboratory.

### 2.2. Transfection

siRNA transfection was performed using Lipofectamine 3000 (L3000075, Thermo Fisher Scientific, Waltham, MA, USA) following the manufacturer’s instruction. mESCs (3.0 × 10^5^) were plated in 6-well plates 24 h prior to transfection. A final concentration of 100 nM siRNA was used. For each gene, two independent siRNA sequences were used to ensure specificity and reproducibility. Fresh culture medium was replaced 6–8 h post-transfection. Cells were harvested 48 or 72 h post-transfection. The sequences of siRNAs are listed in Appendix A.

Plasmids were transfected using PolyJet reagent (SL100688, SignaGen Laboratories, Rockville, MD, USA) following the manufacturer’s instructions.

### 2.3. Genome Editing by CRISPR/Cas9

For gene knock-in (KI), multiple guide RNAs (sgRNAs) targeting genomic regions flanking the transcription start site (TSS) of H2AZ.1 were designed using the CRISPOR online tool (https://crispor.gi.ucsc.edu/, accessed on 16 July 2023) [36]. The sgRNA sequences are listed in the corresponding figure panels. The cleavage efficiency of individual sgRNAs was assessed using an mCherry/GFP reporter assay [37]. The sgRNA exhibiting the highest cleavage efficiency was selected for subsequent KI experiments. Donor templates were constructed with 1000 bp homology arms flanking the targeted KI site. The selected PX459-sgRNA plasmid and donor template were co-transfected into mESCs using PolyJet transfection reagent (SL100688, SignaGen Laboratories, Rockville, MD, USA). Puromycin selection was performed for 48 h, after which the antibiotic was withdrawn to minimize the risk of random Cas9 integration. Individual colonies were manually picked and expanded to approximately 100 cells per clone. Genomic DNA was extracted using the Tiangen Micro Genomic DNA Extraction Kit (DP316, TIANGEN, Beijing, China), and PCR analyses were performed as described in Appendix A. PCR products corresponding to the targeted regions were excised from agarose gels and subjected to Sanger sequencing. Correctly targeted clones were further validated by immunoblotting using antibodies against the inserted epitope tags. The primers used in these experiments are listed in Appendix A.

### 2.4. Genome Editing by Prime Editing

Prime editing (PE) was performed as described [38]. Briefly, pegRNAs were designed using the Prime Design tool (https://www.pridict.it/, accessed on 16 November 2024) based on the intended genomic modifications [39]. Synthesized ePegRNA spacer and PR (PBS and RTT) sequences were cloned into BsaI-linearized ePegRNA plasmids (#174038, Addgene, Watertown, MA, USA) [40] and subsequently co-transfected into target cells along with a modified PEmax plasmid (#174038, Addgene, Watertown, MA, USA). In this modified version of the PEmax plasmid, the hMLH1dn sequence was replaced by La protein [41], and a puromycin resistance gene was inserted downstream, separated by a T2A sequence to allow co-expression. Transfections were performed using Lipofectamine 3000 (L3000075, Thermo Fisher Scientific, Waltham, MA, USA) according to the manufacturer’s instructions. Transfected cells were selected with puromycin for 48 h, after which puromycin was removed. Individual cell clones were manually picked and expanded to approximately 100 cells per clone. Genomic DNA was isolated from these clones using the Micro Genomic DNA Extraction Kit (DP316, TIANGEN, Beijing, China), and PCR was performed to amplify the target regions as indicated in the relevant figure legends. PCR products were purified from agarose gels and verified by Sanger sequencing. Correctly targeted clones were further validated by immunoblotting using antibodies against the introduced epitope tags. The relevant primers are listed in Appendix A.

### 2.5. CUT & Tag

The CUT & Tag assay was performed as previously described using the Hyperactive Universal CUT & Tag Assay Kit for Illumina (TD904, Vazyme Biotech, Nanjing, China) [42]. Briefly, 1 × 10^5^ cells were washed with 500 μL of wash buffer and centrifuged at 600× *g* for 5 min at room temperature. The cell pellets were then resuspended in 100 μL of wash buffer. Concanavalin A-coated magnetic beads (10 μL) were washed twice with 100 μL of binding buffer, added to the cell suspension, and incubated at room temperature for 15 min. After removing the supernatant, the bead-bound cells were resuspended in 50 μL of antibody buffer containing primary antibodies against Flag at a 1: 100 dilution. The samples were incubated overnight at 4 °C.

Following incubation, the primary antibody was carefully removed, and the pellets were washed three times with Dig-wash buffer. The samples were then incubated for 1 h at room temperature with rotation in 50 μL of Dig-wash buffer containing secondary anti-rabbit IgG (Ab6702, Abcam, Cambridge, UK). After three gentle washes with 200 μL of Dig-wash buffer, 2 μL of pA/G-Tnp along with 98 μL of Dig-300 buffer was added. Following a 1 h incubation at room temperature, the samples were washed three times with 200 μL of Dig-300 buffer. Subsequently, 10 μL of 5× TTBL mixed with 40 μL of Dig-300 buffer was added to each sample, and the mixtures were incubated at 37 °C for 1 h. To quench the interactions, 5 μL of 20 mg/mL Proteinase K, 100 μL of Buffer L/B, 1 pg spike in, and 20 μL of DNA extraction beads were added, followed by incubation at 55 °C for 10 min. The supernatant was discarded, and the beads were washed once with 200 μL of Buffer WA and twice with 200 μL of Buffer WB, before being resuspended in 22 μL of nuclease-free water.

For library amplification, 15 μL of purified DNA was combined with 25 μL of 2× CAM and 5 μL of uniquely barcoded i5 and i7 primers from the TruePrep Index Kit V2 for Illumina (TD202, Vazyme Biotech, Nanjing, China), resulting in a total reaction volume of 50 μL. The PCR amplification was carried out in a thermal cycler under the following conditions: 72 °C for 3 min; 98 °C for 3 min; 12–16 cycles of 98 °C for 10 s, 60 °C for 5 s, and 72 °C for 1 min; and a final hold at 12 °C. To purify the PCR products, 2× volumes of VAHTS DNA Clean Beads (N411, Vazyme Biotech, Nanjing, China) were added and incubated at room temperature for 5 min. The beads were then washed twice with 200 μL of fresh 80% ethanol and eluted in 22 μL of nuclease-free water. All CUT & Tag libraries were sequenced by Novogene using the Illumina platform in PE150 mode ( Novogene Co., Ltd., Sacramento, CA, USA).

### 2.6. Cut & Tag Data Analysis

For Cut & Tag data, the read quality was checked with FastQC, and preprocessing to remove adapter contamination, low-quality reads, and bases following the same protocol as above. High-quality reads were mapped to the mm10 mouse genome assembly with the STAR v2.7.10b aligner using the following parameters: --outFilterMultimapScoreRange 2 --winAnchorMultimapNmax 1000 --alignIntronMax 1 --alignMatesGapMax 1000 --outFilterMultimapNmax 10000 --outFilterMismatchNmax 10 --outReadsUnmapped Fastx --alignEndsType Local --outSAMprimaryFlag AllBestScore. Duplicate reads were marked and excluded from downstream analyses. Enriched regions were identified using SEACR v1.3 according to the recommended workflow (https://github.com/FredHutch/SEACR, accessed on 16 November 2024), using fragment size-filtered reads to generate bedgraph files as the input for peak calling (0.01 norm stringent). Peak lists were aggregated across replicates using Bedtools v2.29.2 to create a merged peak list, which was then annotated with Homer v4.11. Heatmaps were generated in Deeptools v3.5.0 using normalized input files.

### 2.7. Western Blotting

Whole-cell extracts were prepared by dissolving cells in a denaturing buffer (20 mM Tris, pH 7.4, 50 mM NaCl, 0.5% Nonidet P-40, 0.5% deoxycholate, 0.5% SDS, 1 mM EDTA) supplemented with a protease inhibitor cocktail (PIA32955, Fisher Scientific, Waltham, MA, USA) with sonication. Protein concentration was quantified using the BCA protein assay kit (5000111, Bio-Rad, Hercules, CA, USA). Cell lysates (20 μg) were loaded onto an SDS-PAGE gel and transferred to a nitrocellulose membrane (1620115, Bio-Rad, Hercules, CA, USA). The membrane was blocked with 5% (*w*/*v*) Blotting-grade blocker (1706404, Bio-Rad, Hercules, CA, USA) in TBST [20 mM Tris-HCl, pH 7.6, 150 mM NaCl, 0.1% (*w*/*v*) Tween-20] and then incubated with the primary antibodies overnight at 4 °C, followed by HRP-conjugated secondary antibodies at room temperature for 1 h. Detection was performed using Thermo Scientific™ SuperSignal™ West Pico PLUS Chemiluminescent Substrate (PI34578, Thermo Scientific, Waltham, MA, USA). The antibodies included anti-HA (1:10,000, Proteintech, 51064–2-AP), anti-H3 (1:5000, Abcam, 1791), anti-H4 (1:2000, Proteintech, 16047–1-AP), anti-H2AZ (1:2000, Abclonal, A4599), anti-Flag (1:5000, Proteintech, 20543–1-AP), anti-Flag (1:500, Biolegend, 637302), anti-GAPDH (1:5000, Proteintech, 10494–1-AP), anti-BAP1 (1:2000, active motif, 61502), anti-Ring1B (1:2000, Cell Signaling Technology, 5694), anti-H2AK119ub (1:2000, Cell Signaling Technology, 8240), anti-USP16, and anti-Ring 1A antibodies, which have been previously described [43,44]. All experiments were performed at least three times independently to ensure reproducibility.

### 2.8. Nucleosome Isolation and in Vitro Histone Ubiquitin Ligase Assay

Nucleosomes were purified from Flag-H2AZ.1 or Flag-H2AZ.2 293T cells, following a previously described protocol [45]. Briefly, 2 × 10^8^ cells were digested with 20 U of micrococcal nuclease (LS004797, Worthington Biochemical Corporation, Lakewood, NJ, USA) for 1 h to obtain mono-nucleosomes. Nucleosomes were extracted and then subjected to Sephacryl S-300 (17-1167-01, Cytiva, Marlborough, MA, USA) gel filtration column purification. The protein and DNA profiles of Sephacryl S-300 were analyzed by Coomassie brilliant blue (CBB) staining and agarose gel electrophoresis, respectively [45]. The mono-nucleosome or oligo-nucleosome fractions were pooled respectively, dialyzed against histone storage buffer (HSB, 10 mM Hepes-KOH pH 7.5, 1 mM EDTA, 60 mM KCl, 0.2% NP-40, 10% glycerol, 0.2 mM PMSF). The nucleosomes were then subjected to immunoprecipitation by anti-Flag M2 agarose (A2220, Sigma-Aldrich, St. Louis, MO, USA), and bound nucleosomes were eluted with Flag peptides. To assess the impact of pre-existing H2A ubiquitination, a portion of the purified nucleosomes was treated with USP16 protein (purified from Sf9 insect cells) to remove H2AK119ub. Both untreated and USP16-treated nucleosomes were used as substrates in the ubiquitination reactions.

For the in vitro histone ubiquitin ligase assay, 5 μg of mono-nucleosomes or oligo-nucleosomes were used in a 36 μL reaction system containing 50 mM Tris-HCl (pH 7.9), 5 mM MgCl_2_, 2 mM NaF, 0.6 mM DTT, 0.6 mM ATP, 10 μM Okadaic acid, 0.1 μg ubiquitin activating enzyme E1 (E-304-050, R&D Systems, Minneapolis, MN, USA), 0.6 μg ubiquitin conjugating enzyme Ubc5c, 1 μg HA-tagged ubiquitin (U-110, Boston Biochem, Cambridge, MA, USA), and 0.6 μg E3 ligase [46]. Reactions were incubated at 37 °C for 30 min and terminated by the addition of SDS-PAGE loading buffer. Proteins were separated by 18% SDS-PAGE and analyzed by immunoblotting with anti-HA antibody (51064-2-AP, Proteintech, Rosemont, IL, USA). All assays were performed at least in triplicate to ensure reproducibility.

### 2.9. Co-Immunoprecipitation Experiment

A total of 1.0 × 10^7^ mESCs expressing HA-H2AZ.1 and Flag-H2AZ.2 were lysed in 800 μL of IP-TNE buffer (10 mM Tris-HCl, 150 mM NaCl, 1 mM EDTA, 1% NP-40, pH 7.5) at 4 °C for 30 min. Lysates were centrifuged at 16,000× *g* 4 °C for 30 min; the resulting pellet was resuspended in 800 μL buffer of fresh IP-TNE buffer. The suspension was then sonicated for 3 min using a cycle of 3 s on and 5 s off at 30% output QSONICA sonicator (Q55, Newtown, CT, USA). Following sonication, the samples were centrifuged at 4 °C at 800× *g* for 8 min. The supernatant was collected and used for immunoprecipitations with either anti-Flag (M8823-MB, Sigma-Aldrich, St. Louis, MO, USA) or anti-HA (88836, Thermo Scientific, Rockford, IL, USA) magnetic beads. After incubation at 4 °C for 2 h, the beads were washed with IP-TNE buffer and eluted using Flag or HA peptides at a final concentration of 0.5 mg/mL for 2 h. Aliquots of input, flowthrough, and eluate fractions were analyzed by Western blotting.

### 2.10. Statistical Analysis

Quantitative data are presented as mean ± SEM from at least three independent experiments. Statistical significance was assessed using Student’s *t*-test for comparisons between two groups, and one-way ANOVA for comparisons among three or more groups. The Chi-squared test was used for categorical data where appropriate. *p* < 0.05 was considered statistically significant.

## 3. Results

### 3.1. The Two Variants of Histone H2AZ, H2AZ.1 and H2AZ.2, Are Differentially Expressed in Mouse Embryonic Stem Cells

The two variants of histone H2AZ, H2AZ.1 and H2AZ.2, share identical protein sequences between human and mouse and differ by only three amino acids, threonine (T), serine (S), and valine (V), in H2AZ.1 vs. alanine (A), threonine (T), and alanine (A) at positions 15, 39, and 128 (Figure 1A). Due to this property, currently available antibodies cannot distinguish between these isoforms, making it difficult to study their individual biological functions. To address this limitation, we knocked a Flag epitope tag into the 5′ end of endogenous *H2afz* loci (encoding H2AZ.1) in mESCs using the CRISPR/Cas9 genome editing system (Appendix A). This knock-in cell line, together with the recently reported Flag-H2AZ.2 KI mESC line [35], enable a direct comparison of the endogenous levels of the two variants in mESCs. Both cell lines expressed H2AZ.1 and H2AZ.2 at levels comparable to their endogenous counterparts, indicating that Flag tagging did not affect their expression (Figure 1B). Immunoblot analyses using anti-Flag antibody revealed a substantial difference in the protein levels of the two isoforms (Figure 1C, lane 1–3, marked with an asterisk). Protein titration and quantification showed that H2AZ.2 protein levels were approximately one sixth those of H2AZ.1, consistent with previous studies in human cells [31] (Appendix A). Notably, we also observed an additional band migrating at approximately 23 kDa (Figure 1C, marked with a pound), likely representing monoubiquitinated forms of H2AZ.1 or H2AZ.2. These modified forms represent roughly 10% of the non-modified form, suggesting that approximately one tenth of H2AZ.1 and H2AZ.2 is monoubiquitinated in mESCs.

### 3.2. Both H2AZ.1 and H2AZ.2 Regulate H2AK119ub Levels

In a recent study, we identified H2AZ.2 as a component of H2AK119ub-modified nucleosomes and demonstrated that its knock-out impaired PRC1-mediated H2AK119ub [35]. To investigate whether the closely related variant H2AZ.1 also regulates this modification, we attempted to knockdown this variant specifically. To exclude the potential effects of the knockdown one variant on the other (see below), we generated double knock-in (DKI) mouse ESCs expressing HA-tagged H2AZ.1 and Flag-tagged H2AZ.2 using prime editing technology (Appendix A). Upon siRNA-mediated knockdown (KD), we found that while H2AZ.2 siRNA specifically reduced H2AZ.2 protein levels, H2AZ.1 siRNA not only decreased H2AZ.1 itself but also reduced H2AZ.2 protein levels (Figure 2A, 1st and 2nd panels; compare lane 2 with 3). RT-qPCR results revealed that the knockdown of H2AZ.1 also led to a reduction in H2AZ.2 mRNA levels (Figure 2E), suggesting the existence of a unidirectional regulatory relationship between H2AZ.1 and H2AZ.2. Consistent with previous reports, the knockdown of H2AZ.2 led to a reduction in H2AK119ub levels (Figure 2A, 3rd panel; compare lane 2 with 3). Notably, H2AZ.1 depletion caused a more pronounced decrease in H2AK119ub levels, unlikely to be due to the concurrent downregulation of H2AZ.2 (Figure 2A, top three panels; compare lane 2 with 3), since the extent of H2AZ.2 reduction is less pronounced in H2AZ.1 KD cells (Figure 2A; compare lane 2 with 3 in the 1st and 3rd panels). These findings suggest that both H2AZ.1 and H2AZ.2 regulate H2AK119ub levels and may function through distinct mechanisms.

To investigate the mechanisms underlying these regulations, we measured the levels of PRC1 core subunits Ring1A and Ring1B. The knockdown of H2AZ.1 led to a significant reduction in Ring1A and Ring1B protein levels, whereas the knockdown of H2AZ.2 had no observable effect on the protein levels of Ring1A and Ring1B (Figure 2C, top two panels; compare lanes 2 with 3, quantified in panel D). RT-qPCR analysis revealed no significant changes in Ring1A or Ring1B mRNA levels, indicating that the regulation occurs at post-transcriptional, translational, or protein stability levels (Figure 2E). Moreover, the knockdown of either variant did not affect the levels of reported H2A deubiquitinases USP16 and BAP1, ruling out the possibility that the effects took place through changes in H2AK119ub deubiquitination (Figure 2C, 3rd and 4th panels; compare lanes 2 with 3).

To determine the potential mutual regulation—specifically, whether PRC1-mediated H2AK119ub influences H2AZ.1 and H2AZ.2 protein levels—we simultaneously knocked down Ring1A and Ring1B in DKI mESCs (Figure 2F, 1st and 2nd panels; compare lane 1 with 2, 3, 4). Dual Ring1A/Ring1B KD, or KD Ring1B KD alone, significantly reduced H2AK119ub levels (Figure 2F, 4th panel; compare lanes 3 and 4 with 1 and 2. The quantification is shown on the right). Western blot analysis revealed that, consistent with the loss of H2AK119ub, the depletion of Ring1B—either individually or in combination with Ring1A—led to a 40–50% reduction in H2AZ.2 protein levels, while H2AZ.1 levels remained unchanged (Figure 2G, 2nd panel; compare lanes 3 and 4 with 1 and 2. The quantification is shown on the right). These findings suggest that although H2AZ.2 and H2AZ.1 can both modulate H2AK119ub levels, H2AK119ub affects the protein levels of these two H2AZ variants differently.

### 3.3. Genomic Distribution and Co-Localization of H2AZ.1, H2AZ.2, and H2AK119ub

To explore the genome-wide distribution and potential functional relationships among histone variants H2AZ.1, H2AZ.2, and H2AK119ub, we performed CUT & Tag assays. Heatmap and metagene analyses revealed similar enrichment profiles for H2AZ.1, H2AZ.2, and H2AK119ub centered around transcription start sites (TSSs), within a ± 5 kb window (Figure 3A). The genomic annotation of peaks showed that H2AZ.1 and H2AZ.2 were distributed similarly across functional genomic regions, with the majority located in intergenic regions (37.12% and 36.56%, respectively), followed by upstream promoter regions (5 kb upstream of TSS; 19.95% for H2AZ.1 and 19.16% for H2AZ.2), downstream regions (12.54% and 12.19%), exons (11.80% and 12.36%), and introns (18.59% and 19.73%) (Figure 3B,C). H2AK119ub displayed a comparable pattern, being slightly more enriched in intergenic regions (34.33%) and more frequently found within intronic regions (30.14%) (Figure 3D).

Co-localization analysis revealed substantial overlap between H2AZ.1 and H2AZ.2-bound genes, with 89.85% of H2AZ.1-bound genes also bound by H2AZ.2, and 86.4% vice versa, suggesting a possible functional redundancy or cooperation between these variants (Figure 3E). Notably, approximately 53% of genes bound by H2AZ.1 or H2AZ.2 also harbored H2AK119ub binding (Figure 3F,G), indicating significant genomic co-localization. While the association of H2AZ.2 with H2AK119ub is consistent with our previous findings [35], the role of H2AZ.1 in regulating H2AK119ub is unexpected. The enrichment analysis of genes co-bound by H2AZ.1, H2AZ.2, and H2AK119ub reveals significant insights into their regulatory roles in development. Genes co-bound by H2AZ.1 and H2AZ.2 are enriched in biological processes such as neurogenesis, axonogenesis, cell fate commitment, and tissue morphogenesis, indicating their shared involvement in developmental programming. Similarly, genes co-bound by H2AZ.1 or H2AZ.2 with H2AK119ub were also enriched in biological processes such as mesenchyme development, synapse organization, and epithelial tissue morphogenesis. These overlaps suggest that the two H2AZ isoforms and H2AK119ub may function cooperatively to regulate chromatin structure and gene expression at key developmental loci (Appendix A).

KEGG pathway analyses further highlighted shared enrichments across multiple signaling pathways critical for stem cell function and differentiation, including the Wnt, MAPK, PI3K-Akt, calcium, and cAMP pathways. These results are consistent with the known role of H2AK119ub as a Polycomb-associated repressive mark and point to a chromatin context in which H2AZ.1 and H2AZ.2 are involved in fine-tuning gene repression marked by H2AK119ub. Notably, there are subtle differences in pathway enrichment between H2AZ.1 and H2AZ.2, particularly for neuronal and synaptic-related processes. Additionally, genes uniquely bound by H2AZ.1 are significantly enriched in biological processes such as ribosome biogenesis, rRNA metabolism, nuclear organization, and cell cycle progression, including autophagy (Appendix A). In contrast, genes exclusively associated with H2AZ.2 show enrichment in pathways related to vesicle localization, phospholipid metabolism, carbohydrate homeostasis, and cytokine-mediated signaling.

These data suggest that both H2AZ variants are enriched at promoters and strongly co-localize with PRC1-mediated H2AK119ub. Since our data show that the knockdown of either H2AZ variant leads to a reduction in H2AK119ub levels (Figure 2), a key question arises: how do these two isoforms influence this histone modification?

### 3.4. H2AZ.1 KD Reduced the Levels of Ring1B on Chromatin

Given the established role of H2AZ.1 in promoting chromatin accessibility [47,48,49], we hypothesize that H2AZ.1 may modulate H2AK119ub levels via facilitating PRC1 recruitment. To delineate the distinct roles of H2AZ.1 and H2AZ.2 in PRC1 recruitment, we performed CUT & Tag assays using a Flag-Ring1B knock-in mESC line following the siRNA-mediated knockdown of H2AZ.1 or H2AZ.2. Heatmap analysis of Ring1B genomic occupancy around TSS ( ± 5 kb) revealed a substantial reduction in Ring1B binding upon H2AZ.1 depletion, while H2AZ.2 knockdown resulted in only a modest effect relative to control cells (Figure 4A). These results suggest that H2AZ.1 plays a primary role in facilitating Ring1B association with chromatin. Further examination of canonical PRC1 target genes (*Hoxa9*, *HoxC5*, and *HoxD10*) and known Ring1B-regulated loci (*Klf2* and *DKK1*) [50] demonstrated a marked decrease in Ring1B occupancy in H2AZ.1-depleted cells, with minimal change observed following H2AZ.2 knockdown (Figure 5B–F). These data highlight distinct yet complementary functions for the H2AZ variants in H2AK119ub regulation. While H2AZ.1 primarily facilitates the recruitment of PRC1 and thus promotes the deposition of H2AK119ub, H2AZ.2 may contribute to the structural maintenance of H2AK119ub-marked nucleosomes. Together, these H2AZ isoforms support a mechanistic framework that fine-tunes PRC1 subunit recruitment and histone ubiquitination, ensuring proper gene silencing by PRC1.

### 3.5. H2AZ.1-Containing Nucleosomes Are Better Substrates for PRC1-Mediated Histone H2A Ubiquitination

To determine whether H2AZ.1- or H2AZ.2-containing nucleosomes are more favorable substrates for histone H2A ubiquitination, we purified nucleosomes containing either H2AZ.1 or H2AZ.2 and performed in vitro ubiquitin ligase assays with PRC1 as the E3 ligase (Appendix A). We first generated 293T cell lines stably expressing Flag-H2AZ.1 or Flag-H2AZ.2 and extracted nucleosomes. The isolated nucleosomes were purified through Sephacryl S-300 gel filtration, which separates mono- and different lengths of oligo-nucleosomes (Appendix A). Fractions corresponding to mono- and oligo-nucleosomes were pooled and subjected to anti-Flag immunoprecipitation. Elution with Flag-peptides yielded nucleosomes enriched for H2AZ.1 or H2AZ.2, as confirmed by immunoblotting (Figure 5A).

To make these nucleosomes better substrates for in vitro histone ubiquitin ligase reaction, we treated these nucleosomes with USP16 to remove H2AK119ub (Figure 5B). The immunoblot results confirmed a marked reduction in H2AK119ub levels following USP16 treatment. In subsequent in vitro ubiquitin ligase assays, both mono- and oligo-nucleosomes containing H2AZ.1 exhibited higher levels of ubiquitination compared to their H2AZ.2 counterparts (Figure 5C,D). This same situation also applied to substrates when H2AK119ub was removed by USP16 treatment (Figure 5E,F). These results indicate that H2AZ.1-containing nucleosomes are preferred substrates for PRC1-mediated H2A ubiquitination.

## 4. Discussion

In this study, we dissect the functional divergence between the two histone H2AZ variants—H2AZ.1 and H2AZ.2—in the context of PRC1-mediated H2AK119ub in mESCs. By leveraging isoform-specific knock-in mESC models, we provide compelling evidence that despite their near-identical amino acid sequences and overlapping genomic distribution, H2AZ.1 and H2AZ.2 execute non-redundant roles in chromatin regulation and Polycomb-mediated histone H2A ubiquitination.

Our genomic and biochemical analyses reveal that H2AZ.1 is a primary facilitator of PRC1 recruitment, promoting Ring1B occupancy at the target loci. This role is consistent with previous studies linking H2A.Z to facilitating the recruitment of chromatin-modifying enzymes, although in these studies, H2AZ.1 and H2AZ.2 were not differentiated [20,49,51,52]. Interestingly, the depletion of H2AZ.1 also significantly reduced H2AZ.2 levels, indicating a potential hierarchical or stabilizing relationship between the isoforms. In contrast, H2AZ.2 did not appear to facilitate PRC1 recruitment, as Ring1B binding was largely unaffected by its knockdown. Instead, H2AZ.2 may contribute structurally, supporting the stability of H2AK119ub-modified nucleosomes, though the exact role of H2AZ.2 in H2AK119ub-modified nucleosomes remains to be determined.

Our in vitro ubiquitination assays further substantiate this division of labor. H2AZ.1-containing nucleosomes were more efficiently ubiquitinated by PRC1 than their H2AZ.2-containing counterparts, regardless of whether pre-existing H2AK119ub was removed. This finding underscores the greater catalytic competency of H2AZ.1 nucleosomes in serving as substrates for PRC1 and complements the chromatin-level evidence for H2AZ.1′s role in promoting Ring1B association. Together, our data support a model in which H2AZ.1 initiates and facilitates PRC1-mediated chromatin modification, while H2AZ.2 contributes to the maintenance and structural reinforcement of the repressive chromatin environment [35,53]. This functional dichotomy may be critical for fine-tuning Polycomb-dependent gene regulation in a dynamic and context-dependent manner.

Our findings also raise several new questions. For instance, the precise structural basis for PRC1’s differential catalytic efficiency toward H2AZ.1 versus H2AZ.2 nucleosomes remains unclear. Future studies utilizing cryo-EM and single-molecule analyses may provide structural insights into how subtle isoform-specific differences influence H2AK119ub differentially [54,55].

## 5. Conclusions

This work defines distinct mechanistic roles for H2AZ.1 and H2AZ.2 in regulating H2AK119ub. By resolving their individual contributions, we establish a functional framework in which histone variant isoforms integrate regulatory complexity into the Polycomb epigenetic network, offering new avenues for understanding chromatin specialization in development and disease.

## Figures and Tables

**Figure 1 cells-14-01133-f001:**
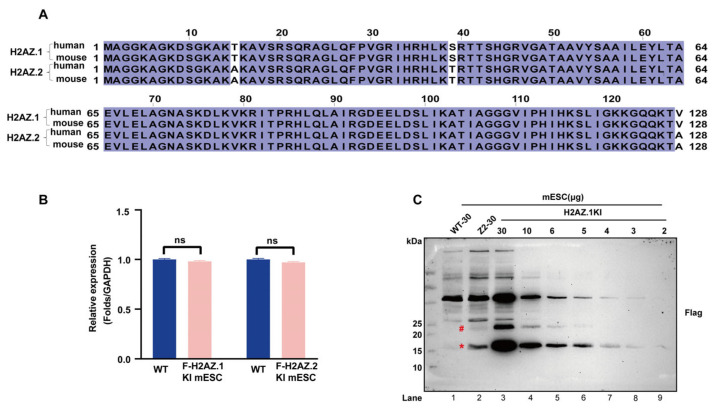
H2AZ.1 and H2AZ.2 are expressed at different levels in mESCs. (**A**) Sequence alignment of human and mouse H2AZ.1 and H2AZ.2 proteins. Conserved residues are highlighted in dark blue. Sequences include human H2AZ.1 (P0C0S5), mouse H2AZ.1 (P0C0S6), human H2AZ.2 (Q71UI9), and mouse H2AZ.2 (Q3THW5). (**B**) RT-qPCR analyses of H2AZ.1 and H2AZ.2 mRNA levels in wild and Flag-tagged knock-in (KI) H2AZ.1 and H2AZ.2 mESCs. (**C**) Immunoblot analysis of the protein levels of Flag-KI-H2AZ.1 and Flag-KI-H2AZ.2 in mESCs. Total proteins were prepared by completely dissolving control (WT), Flag-KI-H2AZ.1, and Flag-KI-H2AZ.2 mESCs with sonication. Serial dilutions (30 to 2 µg) of Flag-KI-H2AZ.1 proteins were loaded. “WT-30” refers to 30 µg of total protein extracted from wild-type mESCs, and “Z2–30” refers to 30 µg of total protein from the Flag-H2AZ.2 knock-in mESC line. A CBB-stained gel is shown in Appendix A for protein loading. Target bands are marked with red and asterisk. Bands likely representing ubiquitinated H2AZ.1 and H2AZ.2 are marked with a pound.

**Figure 2 cells-14-01133-f002:**
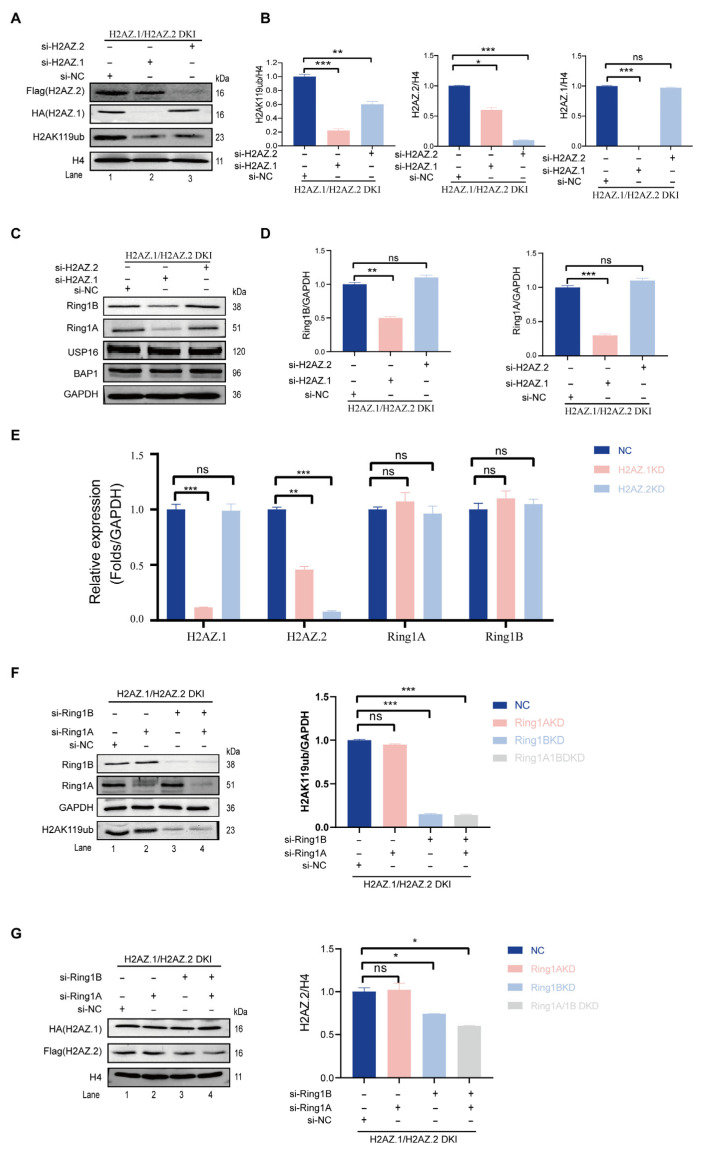
H2AZ.1 and H2AZ.2 both regulate H2AK119ub levels but through distinct mechanisms. (**A**) Immunoblot analysis of H2AZ.1 and H2AZ.2 DKI mESCs transfected with siRNAs targeting H2AZ.1 or H2AZ.2 or non-targeting control (si-NC). H2AK119ub levels were detected using an anti-H2AK119ub antibody. Histone H4 served as the loading control. (**B**) The quantification of H2AK119ub (left), H2AZ.2 (middle), and H2AZ.1 (right) protein levels normalized to histone H4 in DKI cells under siRNA treatment. (**C**) Immunoblots of Ring1A and Ring1B levels in DKI mESCs following the knockdown of H2AZ.1 and/or H2AZ.2. GAPDH was used as the loading control. The same samples used in panel A were analyzed. (**D**) The quantification of Ring1A (**left**) and Ring1B (**right**) levels normalized to GAPDH in DKI mESCs under siRNA treatment. (**E**) RT-qPCR analysis of H2AZ.1, H2AZ.2, Ring1A, and Ring1B mRNA levels normalized to GAPDH in cells treated with siRNAs against H2AZ.1 or H2AZ.2. (**F**) Immunoblot analysis of DKI mESCs transfected with siRNAs against Ring1A, Ring1B, or both. Levels of H2AK119ub were assessed, and quantification is shown on the right. (**G**) Immunoblot analysis of DKI mESCs transfected with siRNAs against Ring1A, Ring1B, or both. Levels of H2AZ.1 and H2AZ.2 are quantified on the right (normalized to H4). The same samples as in panel F were used. Data are presented as mean ± SEM, *n* ≥ 3. Statistical significance: ns, not significant. * *p* < 0.05, ** *p* < 0.01, *** *p* < 0.001.

**Figure 3 cells-14-01133-f003:**
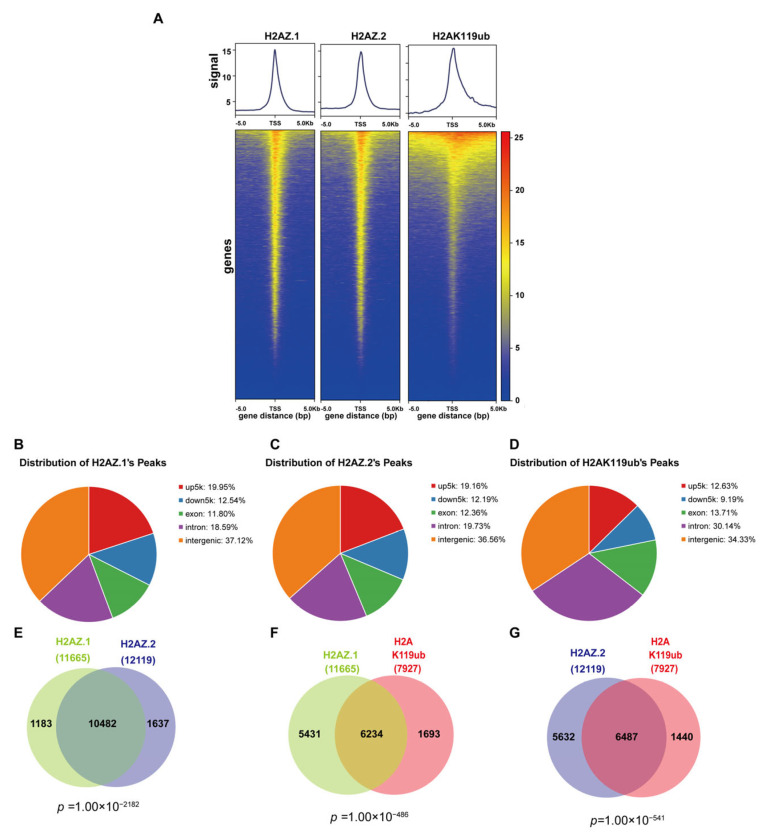
Genomic distribution and co-localization of H2AZ.1, H2AZ.2, and H2AK119ub. (**A**) Metagene profiles (**top**) and heatmaps (**bottom**) show the genome-wide distribution of H2AZ.1, H2AZ.2, and H2AK119ub signals around transcription start sites (TSSs ± 5 kb). Each row in the heatmap represents a gene, ordered by signal intensity. (**B**–**D**) Genomic annotation of peaks for H2AZ.1 (**B**), H2AZ.2 (**C**), and H2AK119ub (**D**). Peaks are classified into promoter regions (upstream 5 kb of TSS), downstream regions (downstream 5 kb of TES), exons, introns, and intergenic regions. (**E**–**G**) Venn diagram showing the overlap between H2AZ.1- and H2AZ.2-bound genes (**E**), H2AZ.1- and H2AK119ub-bound genes (**F**), and H2AZ.2- and H2AK119ub-bound genes (**G**). The *p*-value below the diagram indicates the significance of the intersection, determined by a Chi-squared test with a background of 20,000 genes.

**Figure 4 cells-14-01133-f004:**
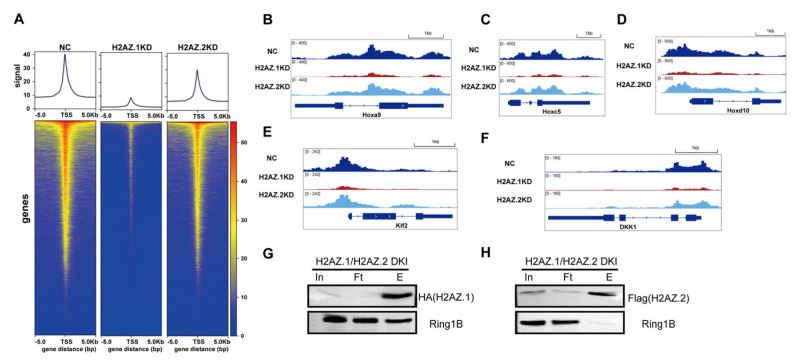
H2AZ.1 but not H2AZ.2 regulates the genomic occupancy of PRC1 subunit Ring1B. (**A**) Metagene profiles (**top**) and heatmaps (**bottom**) of Ring1B CUT & Tag signals around TSSs (±5 kb) in control (NC), H2AZ.1 knockdown, and H2AZ.2 knockdown mESCs. Ring1B occupancy is substantially reduced in H2AZ.1-depleted cells and slightly decreased in H2AZ.2-depleted cells. All CUT & Tag experiments were performed in three biological replicates. The validation of CUT & Tag samples by Western blotting is shown in Appendix A. (**B**–**F**) Genome browser snapshots showing Ring1B occupancy at representative loci: (**B**) *Hoxa9*, (**C**) *Hoxc5*, (**D**) *Hoxd10*, (**E**) *Klf2*, and (**F**) *DKK1*. Tracks show Ring1B enrichment in NC, H2AZ.1 KD, and H2AZ.2 KD samples. H2AZ.1 knockdown causes a more dramatic reduction in Ring1B occupancy than H2AZ.2 knockdown across multiple target genes, consistent with a role for H2AZ.1 in facilitating PRC1 chromatin binding. (**G**,**H**) Ring1B associates with H2AZ.1- but not H2AZ.2-containing nucleosomes. Immunoprecipitation was performed using anti-HA (**G**) and anti-Flag (**H**) magnetic beads on nucleosome-containing fractions from dual-tagged mESCs expressing HA-H2AZ.1 and Flag-H2AZ.2. The presence of Ring1B was assessed in the input (In), flow-through (Ft), and eluate (E) fractions.

**Figure 5 cells-14-01133-f005:**
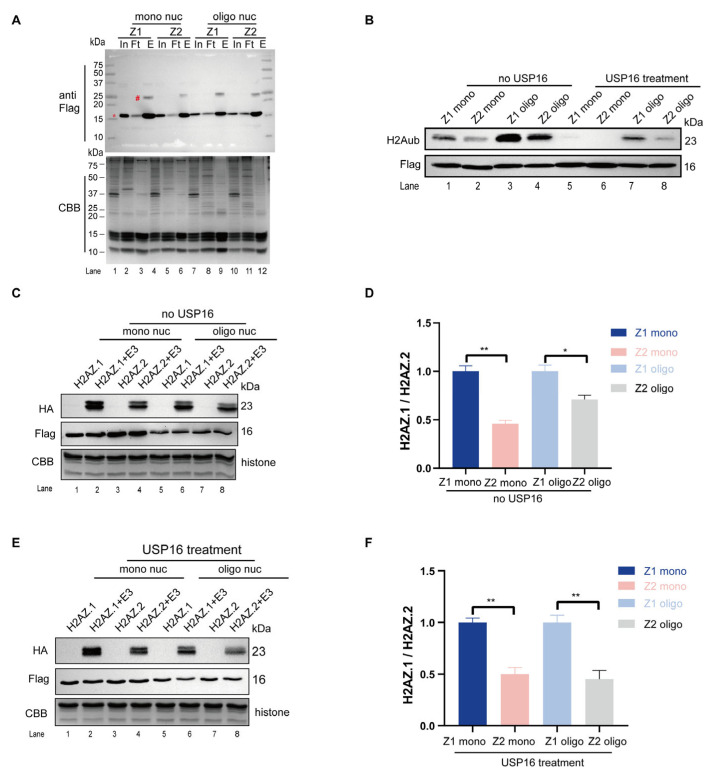
H2AZ.1-containing nucleosomes are preferred substrates for PRC1. (**A**) Immunoblot analysis (**top**) and CBB-stained SDS-PAGE (**bottom**) showing the purification of H2AZ.1- and H2AZ.2-containing mono- and oligo-nucleosomes by anti-Flag immunoprecipitation. Target bands were marked with * while the ubiquitinated bands were marked with #. Input (In), Flowthrough (Ft), and eluent (E) are shown. (**B**) Deubiquitination of purified H2AZ.1- and H2AZ.2-containing mono- and oligo-nucleosomes by USP16. The H2AK119ub level is shown by an anti-H2AK119ub immunoblot (top) and the nucleosome amounts are revealed by anti-Flag immunoblots. (**C**) In vitro ubiquitin ligase assay using H2AZ.1- and H2AZ.2-containing mono- and oligo-nucleosomes (with H2AK119ub not removed by USP16) as substrates and RNF2-Bmi1 as an enzyme. The two bands represent the ubiquitination of endogenous H2A (lower band) and Flag-tagged H2AZ.1 or H2AZ.2 (upper band). Notably, the levels of Flag-tagged H2AZ.1 or H2AZ.2 were significantly less in oligo-nucleosomes compared to mono-nucleosomes when equal amounts of nucleosomes were used. (**D**) The quantification of ubiquitinated H2A signal in panel C. The HA-tagged ubiquitinated H2A (~23 kDa) band was quantified using Image J version 1.53t. The HA signal was normalized to the total Flag-H2AZ nucleosome levels (anti-Flag immunoblotting) to control for input, with the CBB-stained gel serving as a loading control for histone content. (**E**) In vitro ubiquitin ligase assay using H2AZ.1- and H2AZ.2-containing mono- and oligo-nucleosomes (with H2AK119ub removed by USP16) as substrates and RNF2-Bmi1 as an enzyme. (**F**) The quantification of the ubiquitin H2A signal from panel E. Quantification was performed as described in panel D. Data are presented as mean ± SEM, *n* ≥ 3. Statistical significance: * *p* < 0.05, ** *p* < 0.01.

## Data Availability

All high-throughput sequencing data, including H2AZ.1 CUT & Tag and Ring1B CUT & Tag data, have been deposited in the NCBI GEO repository under accession number GSE298579. The H2AZ.2 and H2AK119ub CUT & Tag datasets were obtained from a previously published preprint (bioRxiv, 2024) [35] entitled “Role of histone variants H2BC1 and H2AZ.2 in H2AK119ub nucleosome organization and Polycomb gene silencing”. These data have been deposited in GSE253967.

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
