# Peer review of "Divergent Mechanisms of H2AZ.1 and H2AZ.2 in PRC1-Mediated H2A Ubiquitination"

_cells, 2025, doi:10.3390/cells14151133_

Round 1
Reviewer 1 Report
Comments and Suggestions for Authors
In the submitted manuscript „Divergent Mechanisms of H2AZ.1 and H2AZ.2 in PRC1-Mediated H2A Ubiquitination”, Shen et al. study how H2AZ variants H2AZ.1 and H2AZ.2 regulate PRC1-mediated H2AK119 ubiquitination by utilizing isoform-specific epitope-tagged knock-in mouse embryonic stem cell (ESC).
In order to directly compare two H2AZ isoforms, the authors developed a dual-tagged mESC line expressing HA-tagged H2AZ.1 and FLAG-tagged H2AZ.2.
They report that knockdown of H2AZ.1 reduced H2AZ.2 at both mRNA levels and protein levels. Also, silencing of H2AZ.1 decreased the levels of H2AK119ub and protein levels of PRC1 subunits Ring1A and Ring1B more profoundly than the silencing of H2AZ.2.
Furthermore, depletion of H2AZ.1 (and not H2AZ.2) lead to decreased Ring1B genomic occupancy around TSS, suggesting that H2AZ.1 facilitates Ring1B chromatin association.
Also, depletion of Ring1B lead to decreased H2AK119ub and H2AZ.2 protein levels, with H2AZ.1 levels remaining stable.
These findings are very interesting and points out to the different functions of these 2 isoforms.
However, how can the authors undoubtedly exclude that the effect of H2AZ.1 is not (at least partially or fully) due to off-target effect of H2AZ.1 siRNA (i.e., that it silences both H2AZ.1 and H2AZ.2)? Please provide clear evidence that there are no off-target effects of H2AZ.1 siRNA.
The authors suggest that H2AZ.1 and H2AZ.2 regulate H2AK119ub through distinct mechanisms: H2AZ.1 by facilitating PRC1 recruitment, and H2AZ.2 by maintaining the stability of H2AK119ub-marked nucleosomes.
How do the authors propose that H2AZ.2 regulates H2AK119ub by maintaining the stability of H2AK119ub-marked nucleosomes and why is H2AZ.1 not able to do that?
Is Ring1B able to directly interact with H2AZ.1, opposite to H2AZ.2?
I suggest using dually tagged mESCs to immunoprecipitate tag-bound complexes (followed by either western blot or mass spectrometry analyses).
It is quite remarkable that such subtle sequence differences might account for different functions of H2AZ.1 and H2AZ.2. Are there any available data showing differential phosphorylation of isoform-specific Thr or Ser residues?
Minor comments
Figure 2C: Input for HA- H2AZ.1 and FLAG-H2AZ.2 is missing.
Figure 2G. Quantification of H2AZ.1 normalized to H4 is missing. Only data for H2AZ.2 are provided.
Figure 5D/5F: How exactly has the quantification been performed? What was compared to obtain the quantification?
Author Response
Please see the attachment for our point-by-point response to the reviewer’s comments. Thank you.

Reviewer 2 Report
Comments and Suggestions for Authors
This manuscript focuses on the roles of the histone H2A variant-H2AZ.1 and H2AZ.2 in regulating the H2AK119Ub. The authors knocked-in specific epitope tag to H2AZ.1 and H2AZ.2 in mouse ESC cell lines to dissect the roles of each isoform. Although two isoforms possess highly overlapping genomic binding profiles, the two isoforms play the role via different mechanisms. The research is well-designed, and the data is clearly and reasonably presented.
I have the following concerns: Fig 1A: It seems that the sequence of human and mouse H2AZ.1 and H2AZ.2 is the same. Please confirm it.
Fig 1C. Please explain the sample WT-30 and Z2-30 in the fig legend, it will help the reader understand the Fig 1C. H2AZ is a small protein. It should show a small band shift after it is tagged with Flag. But it is not clear in the image.
Fig 5C and E. Histone ubiquitylation assay. The ubiquitylated H2AZ.1 and H2AZ.2 show 2 bands, what is the possible reason?
Author Response

(The authors gave the same response as above.)
